# Real-World Experience of the Comparative Effectiveness and Safety of Molnupiravir and Nirmatrelvir/Ritonavir in High-Risk Patients with COVID-19 in a Community Setting

**DOI:** 10.3390/v15030811

**Published:** 2023-03-22

**Authors:** Yoshikazu Mutoh, Takumi Umemura, Takeshi Nishikawa, Kaho Kondo, Yuta Nishina, Kazuaki Soejima, Yoichiro Noguchi, Tomohiro Bando, Sho Ota, Tatsuki Shimahara, Shuko Hirota, Satoshi Hagimoto, Reoto Takei, Jun Fukihara, Hajime Sasano, Yasuhiko Yamano, Toshiki Yokoyama, Kensuke Kataoka, Toshiaki Matsuda, Tomoki Kimura, Toshihiko Ichihara, Yasuhiro Kondoh

**Affiliations:** 1Department of Infectious Diseases, Tosei General Hospital, Seto 489-8642, Japan; 2Department of Infection Control Team, Tosei General Hospital, Seto 489-8642, Japan; 3Department of Respiratory Medicine and Allergy, Tosei General Hospital, Seto 489-8642, Japan

**Keywords:** COVID-19, molnupiravir, nirmatrelvir/ritonavir, hospitalization, adverse event, older adults

## Abstract

Molnupiravir (MOV) and nirmatrelvir/ritonavir (NMV/r) are efficacious oral antiviral agents for patients with the 2019 coronavirus (COVID-19). However, little is known about their effectiveness in older adults and those at high risk of disease progression. This retrospective single-center observational study assessed and compared the outcomes of COVID-19 treated with MOV and NMV/r in a real-world community setting. We included patients with confirmed COVID-19 combined with one or more risk factors for disease progression from June to October 2022. Of 283 patients, 79.9% received MOV and 20.1% NMV/r. The mean patient age was 71.7 years, 56.5% were men, and 71.7% had received ≥3 doses of vaccine. COVID-19-related hospitalization (2.8% and 3.5%, respectively; *p* = 0.978) or death (0.4% and 3.5%, respectively; *p* = 0.104) did not differ significantly between the MOV and NMV/r groups. The incidence of adverse events was 2.7% and 5.3%, and the incidence of treatment discontinuation was 2.7% and 5.3% in the MOV and NMV/r groups, respectively. The real-world effectiveness of MOV and NMV/r was similar among older adults and those at high risk of disease progression. The incidence of hospitalization or death was low.

## 1. Introduction

At the end of 2019, the coronavirus (COVID-19) emerged in Wuhan, China. As of February 2023, the COVID-19 pandemic has caused more than 700 million confirmed cases and more than 6 million deaths, with unprecedented changes to global health and social structures worldwide.

The efficacy of remdesivir [1,2], a prodrug of a nucleotide analog inhibiting viral RNA polymerase, was initially established as a treatment for hospitalized patients with COVID-19 requiring oxygen supplementation and was subsequently shown to reduce the risk of hospitalization and death among non-hospitalized patients with COVID-19 who were at high risk for severe disease [3].

In the early stages of the pandemic, various drugs, such as lopinavir/ritonavir, hydroxychloroquine, and favipiravir, were administered; however, they did not provide clear effectiveness [4,5]. Although, for patients with severe COVID-19, immunosuppressive drugs, such as corticosteroids, tocilizumab, and baricitinib, reduced disease progression or death by suppressing the immune response. Anticoagulants, such as heparin and enoxaparin, also played beneficial roles in reducing mortality [6,7]. Subsequently, early treatment with monoclonal antibody therapy was reported to be highly effective in preventing severe disease [8]. Considering that early treatment initiation has the potential to prevent disease progression, molnupiravir (MOV) and nirmatrelvir/ritonavir (NMV/r) were developed as oral antiviral agents.

MOV is a ribonucleoside analog initially designed as a medication for influenza [8].

Once absorbed into the body, it is first metabolized to β-d-N4-hydroxycytidine (NHC), which is then absorbed by cells and converted to its active form, β-d-N4-hydroxycytidine triphosphate (NHC-TP). NHC-TP resembles cytidine triphosphate (CTP) and uridine triphosphate (UTP), which are naturally occurring ribonucleotides used for viral RNA replication. During viral RNA replication, NHC-TP competes with these ribonucleotides and is incorporated into the viral strand for replication by RNA-dependent RNA polymerase, predominantly by SARS-CoV-2. NHC-TP continues to elongate after incorporation into the viral RNA to replicate new viral RNA strands. When the viral RNA strand containing NHC is used as a template for further replication, the NHC-TP in the template behaves as CTP or UTP due to its mutability, thereby creating a replicating fatal error in the new viral RNA (error catastrophe). This error leads to non-infectious viral production and suppression of replication [9,10].

On the other hand, NMV/r selectively inhibits the activity of the main protease of SARS-CoV-2 (Mpro: 3CL protease), which cleaves polyproteins in at least 11 positions to produce nonstructural proteins necessary for viral replication, thus inhibiting viral replication by blocking the cleavage of these polyproteins [11]. Ritonavir (RTV) was originally used as a pharmacokinetic enhancer in the treatment of human immunodeficiency virus infection. Although it does not exhibit antiviral activity against SARS-CoV-2, it inhibits the metabolism of NMV. Thus, RTV improves pharmacokinetics by slowing the metabolism of NMV by inhibiting CYP3A and maintaining the blood concentration of NMV [12].

Both MOV and NMV/r demonstrated good tolerability and safety without any serious adverse events in a clinical trial [8,13]. Additionally, in the MOVe-OUT clinical trial, MOV reduced hospitalization and death through day 29 among non-hospitalized patients with COVID-19 and at least one risk factor for severe illness (7.3% vs. 14.1%; *p* = 0.001) [14]. In the EPIC-HR trial, NMV/r was associated with an 88.9% reduction in COVID-19-related hospitalization and death among patients with COVID-19 at high risk of progression to severe disease [15].

Both MOV and NMV/r have antiviral activity independent of spike protein mutation and are expected to be effective against new mutant strains that may emerge in the future.

As the clinical effectiveness of MOV and NMV/r has been confirmed, these oral antiviral agents were promptly approved for use in many countries. Since the latter part of 2021, the SARS-CoV-2 Omicron variant has emerged and spread rapidly worldwide. The Omicron variant is less likely to cause severe disease than the Delta variant owing to its lower virulence and improved vaccination coverage [16]. Currently, NMV/r is recommended as first-line oral antiviral therapy for COVID-19 in the United States [17]. However, the drug has several contraindications, including concomitant use with certain over-the-counter medications and herbal supplements; therefore, the opportunities for prescription are limited, especially in older adults or patients with other complications. Evidence of the effectiveness of MOV, an alternative antiviral agent against the Omicron variant, is limited. In real-world clinical settings, the optimal oral antiviral treatment strategy for patients with COVID-19 is unclear, especially in older adults and patients at risk of disease progression.

Therefore, we evaluated the comparative effectiveness and safety of MOV and NMV/r in a real-world community setting during the surge of the Omicron BA.5 subvariant in Japan.

## 2. Materials and Methods

We retrospectively evaluated the outcomes of adult patients diagnosed with COVID-19 who were prescribed oral antiviral agents from June to October 2022 in Tosei General Hospital. Either MOV or NMV/r was prescribed for patients with at least one risk factor for progression to severe COVID-19. The risk factors of disease progression were evaluated in accordance with COVID-19 treatment guidelines [7]. NMV/r was prescribed if the patient had no contraindications to the drugs, and the dose was reduced in patients with an estimated glomerular filtration rate of <60 mL/min. MOV was prescribed to patients at risk of disease progression or patients with contraindications for NMV/r. The antiviral drug was switched to remdesivir if the patient required hospitalization unless there were no contraindications to continuing the initial drugs. The diagnosis of COVID-19 was confirmed using Quick Chaser SARS-CoV-2/Flu A,B (Mizuho Medy, Saga, Japan) or Xpert Xpress SARS-CoV-2 (Cepheid, Sunnyvale, CA, USA). Both drugs were initiated within 5 days of symptom onset, and the treatment duration was 5 days. Patient data, including their demographic characteristics, comorbidities, hospitalization history, vaccination status, and clinical course, were collected from their electronic medical records, and a telephone interview was conducted to determine the 28-day outcome.

### 2.1. Definitions

Patients’ medical outcomes were classified as either severe or minor. A severe outcome was defined as COVID-19-related hospitalization or COVID-19-associated mortality within 28 days of symptom onset. COVID-19-related hospitalization was defined as hospitalization because of the symptoms of COVID-19, such as respiratory failure, appetite loss, and pneumonia, and radiological findings. Cases of elective hospitalization, such as those of patients undergoing chemotherapy for malignancy or surgery, after recovery from COVID-19 were excluded. Patients who died within 24 h of treatment initiation were excluded because the treatment compliance was unknown. Patients who were started on oral MOV or NMV/r therapy on admission instead of remdesivir for any reason were excluded from the outcome of hospitalization. Minor outcomes were treatment discontinuation and adverse events. Patients receiving long-term care or support were defined as those living in a long-term care facility, commuting to the facility, or with nurses visiting their homes routinely to administer medical or living care.

### 2.2. Statistical Analysis

The characteristics of patients in the MOV or NMV/r groups were compared using Pearson’s chi-square test or Fisher’s exact for categorical variables and the Mann-Whitney U test for continuous variables. Continuous data are expressed as mean ± standard deviation or median and interquartile range (IQR). Kaplan–Meier survival analysis was performed to compare the cumulative incidence of hospitalization between the MOV and NMV/r groups, and the statistical significance of the differences between groups was assessed using the log-rank test. *p*-values of ≤0.05 were considered statistically significant. Multivariable logistic regression was performed to identify risk factors for hospitalization. The variables were selected according to their clinical relevance. All statistical analyses were performed using SPSS (Version 26.0, IBM Corp, Armonk, NY, USA).

## 3. Results

During the study period, oral antiviral drugs were prescribed to 288 patients; two patients died within 24 h of treatment, and three received delayed prescriptions (>5 days of onset). A total of 283 patients were prescribed MOV (226 patients, 79.9%) or NMV/r (57 patients, 20.1%) (Table 1). More than half the patients were male (56.5%). The mean patient age was 71.7 ± 15.0 years, and 111 (39.2%) were aged over 80 years. Patients in the MOV group were older than those in the NMV/r group; however, this difference was not statistically significant. One-fourth of the patients in each group used long-term care support. Patients in the MOV group were significantly more likely to have a performance status >2 (MOV: 27.0% vs. NMV/r: 12.3%; *p* = 0.01) and significantly more likely to have received four doses of COVID-19 vaccine (MOV: 31.4% vs. NMV/r: 12.3%; *p* = 0.004) than those in the NMV/r group. Nonetheless, the proportion of unvaccinated patients was similar in both groups. The most frequent comorbidity was diabetes (30.7%), followed by chronic heart disease (24.0%), and chronic pulmonary disease (20.8%). Patients in the MOV group were significantly more likely to have two or more risk factors for disease progression than those in the NMV/r group (50.9% vs. 29.8%; *p* = 0.004).

The median time from symptom onset to treatment initiation was 1 day (IQR: 0–2 days) in the MOV group and 1 day (IQR: 0–1 days) in the NMV/r group (*p* = 0.04).

In the MOV group, nine (4.0%) patients received treatment on admission because of contraindications for remdesivir use or chronic kidney failure. Excluding these nine patients, the rate of hospitalization within 28 days was 15/217 (6.9%) in the MOV group and 4/57 (7.0%) in the NMV/r group. The rate of COVID-19-related hospitalization was 6/217 (2.8%) in the MOV group and 2/57 (3.5%) in the NMV/r group (*p* = 0.77). Among the hospitalized patients, corticosteroid, tocilizumab, and heparin were administered to one patient. No patients received monoclonal antibody therapy. The all-cause mortality was 1/226 (0.4%) in the MOV group and 2/57 (3.5%) in the NMV/r group (*p* = 0.1) (Table 2). No significant differences were found in the cumulative hospitalization rate between the two groups (Figure 1). In multivariable logistic regression analysis, the performance status on arrival was associated with the risk of hospitalization, but the selection of the antiviral agent was not (Figure 2). Of the eight patients with COVID-19-related hospitalization, only one patient was admitted because of disease progression and respiratory failure. The other patients were admitted because of other problems, such as general fatigue, throat pain, and loss of appetite. None of the hospitalizations of more than 9 days after symptom onset were related to COVID-19; most were elective hospitalizations for chemotherapy, surgery, or intravascular treatment.

Regarding the cause of death, one patient in the MOV group died of terminal gastric cancer on day 6, and two patients in the NMV/r group died of suspected acute cardiac events in their homes after treatment completion without requiring hospitalization (days 8 and 24).

Overall, 274 (96.8%) patients completed treatment. Adverse events were reported in six (2.7%) patients in the MOV group and three (5.3%) patients in the NMV/r group (*p* = 0.264). The adverse events reported in the MOV group were general fatigue (two patients) and nausea, rash, throat pain, and diarrhea (one patient each). The adverse events reported in the NMV/r group were nausea, general fatigue, and exacerbation of preexisting interstitial pneumonia (one patient each). In addition, three (1.3%) patients in the MOV group and two (3.5%) in NMV/r group discontinued treatment and were hospitalized. (Table 2). All hospitalized patients switched to remdesivir. There were no serious adverse events reported in either group.

## 4. Discussion

In this single-center retrospective study, we showed a real-world situation and the medical outcome of patients at high risk of progressing to severe disease, including adults of advanced age, receiving oral antiviral agents used to treat COVID-19, and we compared the effectiveness and safety of MOV with those of NMV/r. MOV was prescribed four times more frequently than NMV/r, but the risk of disease progression and death within 28 days of symptom onset did not differ significantly according to the choice of antiviral agent. Furthermore, the cause of hospitalization in most cases was not related to disease progression but was associated with social vulnerability based on poor performance status. The incidence of adverse drug events and treatment discontinuation were also similar between the groups. Oral antiviral treatments in the early phase of COVID-19 are comparable, especially in older adults and those with risk factors for disease progression, regardless of the choice of drug.

To date, Japan has experienced eight COVID-19 surges. This study was conducted during the seventh surge of COVID-19, when the Omicron BA.5 subvariant was predominant. To the best of our knowledge, this is the first study to examine real-world medical outcomes of MOV and NMV/r treatment in a community hospital setting during the Omicron BA.5 surge in Japan. MOV and NMV/r were approved at the end of 2021 and the beginning of 2022, respectively, by the Ministry of Health, Labor, and Welfare for use in Japan using a system of special emergency approval.

The efficacy and safety of these drugs against previous variants have been reported in clinical settings [17]. Additionally, the susceptibility of the Omicron variant to both drugs potently inhibited infection [18]. Use of MOV and NMV/r in outpatient or hospitalized patients with COVID-19 has shown favorable outcomes against the Omicron variant, and both drugs are associated with a significantly lower risk of disease progression [19,20,21,22].

In clinical trials, NMV/r has shown higher efficacy than MOV and is the first-line oral COVID-19 treatment according to the US and UK guidelines [23]. However, no direct comparative studies have been conducted to date. NMV/r is contraindicated for co-administration with several drugs, including antiarrhythmic, anticancer, and anticonvulsant drugs, owing to its highly dependent CYP3A4 clearance, and dose adjustment is needed in patients with kidney disorders. Nicolas et al. [24] reported that 14.6% of hospitalized patients with COVID-19 had medical contraindications to NMV/r and that contraindications were more prevalent in men, older patients, and patients with comorbidities. Despite its high level of effectiveness, there is concern about the limited proportion of patients with a high risk of disease progression that could be treated with NMV/r. On the other hand, MOV does not require an adjustment based on renal function, but its administration to pregnant women is contraindicated because of concerns about teratogenicity. There are also concerns about the emergence of new mutant strains and the long-term risk of mutagenicity in humans [25].

Recently, the PANORAMIC study concluded that MOV did not decrease the rate of hospitalization and death but shortened the time to recovery from symptom onset compared with placebo [26]. However, the mean age of participants in the PANORAMIC study was 56.6 years. Since the emergence of the Omicron variant, the primary population of patients hospitalized with COVID-19 is older adults with poor performance status or those who are immunocompromised. Andrea et al. [27] reported that 10.4% of patients with COVID-19 who were treated with MOV experienced disease progression and older age was a risk factor for progression. The PANORAMIC study showed that 1.0% of patients treated with MOV experienced hospitalization or death within 28 days. In our study, the incidence of hospitalization or death within 28 days was 2.8%; however, most of these patients were older adults, and approximately 40% were aged over 80 years. Although a simple comparison is not suitable, considering that most patients hospitalized with the Omicron variant are older adults, MOV is highly effective for older adults and other individuals at high risk of progression who are not eligible for NMV/r. In our study, although the MOV group had more comorbidities and poorer performance statuses than the NMV/r group, effectiveness and safety were similar in both groups. In addition, there was no relationship between treatment response and types of comorbidities between the two drugs. Overall, the initial performance status, but not the choice of drug, was associated with COVID-19-related hospitalization.

Furthermore, the main reason for hospitalization was not disease progression but other symptoms, such as general malaise and appetite loss. Until the emergence of the Delta variant, 15–50% of older adults with COVID-19 progressed to severe disease, and 1 in 10 patients died [28,29]. In contrast, the mortality and disease progression rate related to COVID-19 has decreased dramatically since the emergence of the Omicron variant, even among older adults [30,31,32]. In addition to the early administration of an antiviral agent to patients at high risk of severe disease, the expansion of healthcare resources for socially vulnerable groups is critical for the management of COVID-19.

Regarding adverse events, 96.8% of the patients completed the treatment without experiencing any adverse events. In previous studies, gastrointestinal symptoms, such as nausea, diarrhea, and vomiting, were commonly reported as adverse events of both MOV and NMV/r [33,34]. Rash, dizziness, and headache were less commonly reported adverse events [35]. In our study, the rate of adverse events was 2.7% and 5.3% in the MOV and NMV/r groups, respectively. Although nine patients required discontinuation of treatment, their symptoms rapidly resolved after discontinuing treatment. No serious adverse events were reported. Recently, recurrent symptoms have been reported after treatment in patients treated with MOV or NMV/r, and this issue is thought to be related to viral rebound [36,37,38]. However, in our study, cases of relapse of symptoms or transmission of infection to others after completing treatment were reported on the follow-up telephone call or during follow-up visits in the outpatient department. Although we did not perform polymerase chain reaction or antigen tests after the patients recovered, no patients reported recurrent symptoms. Therefore, there was no reason for concern regarding a possible viral rebound.

Our study has some limitations. First, the decision to hospitalize patients depended on the attending physicians. The seventh COVID-19 wave in Japan was the largest to date, and more than 10% of the population was confirmed COVID-19 positive during the 4-month study period; therefore, some patients could not be hospitalized due to a shortage of hospital beds. However, all patients requiring hospitalization could be hospitalized in our hospital during this period. Second, we do not have data on patients with COVID-19 who were not prescribed antivirals. Since the emergence of the Omicron variant, owing to increased vaccination coverage and lower virulence, COVID-19-related mortality has greatly decreased in all age groups compared with the mortality rate during the Delta predominant period [16]. The effectiveness and safety of MOV and NMV/r were comparable; however, the need for oral antiviral medication was not assessed. Third, this was a single-center retrospective study. MOV was more likely to be prescribed to older patients, those with poor performance status, and those with comorbidities. In contrast, patients who were prescribed NMV/r usually received fewer vaccine doses. Currently, NMV/r is considered more efficacious than MOV; therefore, the United States treatment guidelines recommend NMV/r as the first choice of antiviral drug [16]. Patients who receive MOV were more likely to be older adults because NMV/r is contraindicated in patients with impaired kidney function. Grace et al. [36] noted that the mortality rate of patients treated with MOV was higher than those with NMV/r because MOV was more likely than NMV/r to be prescribed to frail patients with multiple comorbidities and those receiving multiple drugs. A robust comparison of these drugs among patients with the same background is required.

## 5. Conclusions

We compared the effectiveness and safety of MOV and NMV/r in patients with COVID-19, primarily composed of older adults. The MOV group had poorer performance status, was older, and had more comorbidities than the NMV/r group; nevertheless, the rates of hospitalization, death, and adverse events were similar between the two groups. Although either of these antiviral agents is an effective treatment for COVID-19 in those with a risk of progression to severe disease, expansion of healthcare resources is also essential. To establish the appropriate use of antiviral agents for patients with COVID-19, a large prospective study is warranted.

## Figures and Tables

**Figure 1 viruses-15-00811-f001:**
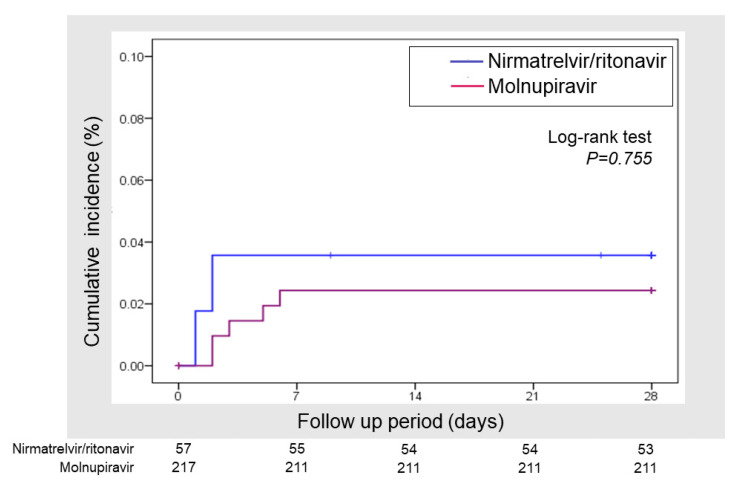
Cumulative hospitalization rate according to the drug type.

**Figure 2 viruses-15-00811-f002:**
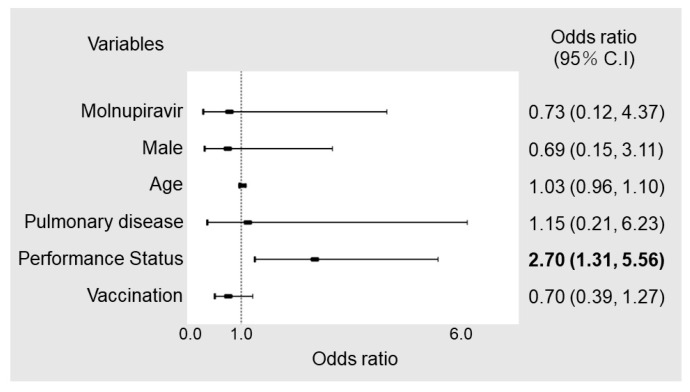
Forest plots of risk factors for hospitalization. Odds ratios were calculated using multivariable logistic regression. C.I., confidence interval.

**Table 1 viruses-15-00811-t001:** Patient characteristics at the time of treatment initiation according to the drug type.

Variables	Total n = 283	MOV n = 226	NMV/r n = 57	*p*-Value
**Age, years (mean ± SD)**	71.7 (±15.0)	72.6 (±14.3)	68.2 (±17.2)	*0.11*
**Male**	160 (56.5%)	127 (56.2%)	33 (57.9%)	*0.47*
**LTCS use**	72 (25.4%)	58 (25.7%)	14 (24.6%)	*0.51*
**Smoker**	20 (7.1%)	12 (5.3%)	8 (14.0%)	*0.03*
**Performance status**				
-0	59 (20.8%)	48 (21.2%)	11 (19.3%)	0.45
-1	156 (55.1%)	117 (51.8%)	39 (68.4%)	0.02
-2	43 (15.2%)	39 (17.3%)	4 (7.0%)	0.04
-3	18 (6.4%)	17 (7.5%)	1 (1.8%)	0.09
-4	7 (2.5%)	5 (2.2%)	2 (3.5%)	0.43
≥2	68 (24.0%)	61 (27.0%)	7 (12.3%)	0.01
**Number of Vaccination**				
-0	39 (13.8%)	28 (12.4%)	11 (19.3%)	*0.20*
-1	2 (0.7%)	0 (0.0%)	2 (3.5%)	*0.04*
-2	39 (13.8%)	31 (13.7%)	8 (14.0%)	*0.95*
-3	(44.2%)	96 (42.5%)	29 (50.9%)	*0.25*
-4	78 (27.6%)	71 (31.4%)	7 (12.3%)	*<0.01*
**Days of treatment initiation from symptom onset, median (IQR)**	1 (0–1)	1 (0–2)	1 (0–1)	*0.04*
**Underlying diseases**				
-Diabetes	87 (30.7%)	74 (32.7%)	13 (22.8%)	*0.10*
-Chronic heart diseases	68 (24.0%)	58 (25.7%)	10 (17.5%)	*0.13*
-Chronic pulmonary diseases	59 (20.8%)	44 (19.5%)	15 (26.3%)	*0.17*
-Central nervous disorder	59 (20.8%)	45 (20.0%)	14 (24.6%)	*0.27*
-Hypertension	56 (19.8%)	46 (20.4%)	10 (17.5%)	*0.39*
-Malignancy	53 (18.7%)	45 (20.0%)	8 (14.0%)	*0.21*
-Chronic kidney diseases	25 (8.8%)	21 (9.3%)	4 (7.0%)	*0.41*
-Hemodialysis	4 (1.4%)	4 (1.8%)	0 (0.0%)	*0.41*
-Obesity (BMI > 30 kg/m^2^)	3 (1.1%)	2 (0.9%)	1 (1.8%)	*0.49*
**≥2 risk factors with disease progression**	132 (46.6%)	115 (50.9%)	(29.8%)	*<0.01*

BMI, body mass index; IQR, interquartile range; LTCS, long-term care support; MOV, molnupiravir; NMV/r, nirmatrelvir/ritonavir; SD, standard deviation.

**Table 2 viruses-15-00811-t002:** Incidence of hospitalization, death, and adverse events according to the drug type.

Variables	MOV n = 226	NMV/r n = 57	*p*-Value
**Hospitalization within 28 days**	15 (6.9%)	4 (7.0%)	*0.98*
-COVID-19-related hospitalization ^†^	6 (2.8%)	2 (3.5%)	*0.77*
**Time to hospitalization from treatment, days ^†^**	3.5 (2–5.5)	1.5 (N/A)	
**Death within 28 days**	1 (0.4%)	2 (3.5%)	*0.10*

**Adverse events**	6 (2.7%)	3 (5.3%)	*0.39*
-General malaise	2 (0.9%)	1 (1.8%)	0.49
-Nausea	1 (0.4%)	1 (1.8%)	0.36
-Rash	1 (0.4%)	0 (0.0%)	>0.99
-Diarrhea	1 (0.4%)	0 (0.0%)	>0.99
-Interstitial pneumonia	0 (0.0%)	1 (1.8%)	0.20
**Treatment cessation**	6 (2.7%)	3 (5.3%)	*0.39*

^†^ Nine patients in the MOV group were excluded; MOV, molnupiravir; NMV/r, nirmatrelvir/ritonavir.

## Data Availability

All data generated or analyzed during this study are included in this published article.

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
