# Peer review of "Real-World Experience of the Comparative Effectiveness and Safety of Molnupiravir and Nirmatrelvir/Ritonavir in High-Risk Patients with COVID-19 in a Community Setting"

_viruses, 2023, doi:10.3390/v15030811_

Round 1

Reviewer 1 Report

The study entitled “Real-world experience of the comparative effectiveness and safety of molnupiravir and nirmatrelvir/ritonavir in high-risk patients with COVID-19 in a community setting” is a very important study. Authors analyzed the role of MOV and NMR/r inhibitors in covid positive patients with older age and most of them have comorbidities. Study is retrospective but important data is generated in this study.

Study is publishable after incorporation of the following comments.

Abstract

Results are the key portion of abstract. Authors describe the vaccination status, adverse effects and age groups in the results. But the true results are the response of MOV and NMV/r. Write in numbers, what is the response of MOV and NMV/r in covid patients with older age.

Introduction is short.

Write a paragraph about the Molnupiravir and nirmatrelvir/ritonavir. What kind of drugs these are? how much effectiveness they showed in clinical trials? what is the response of these drugs in normal adult population?

Write a paragraph about different drugs, which were used at different stages during the covid pandemic for the treatment of covid. You may include Anticoagulants, Steroids etc.

https://pubmed.ncbi.nlm.nih.gov/34827332/

https://pubmed.ncbi.nlm.nih.gov/34943722/

Write a paragraph about use of different drugs in covid positive special populations like the older population, population with impaired kidney. You may cite the papers like Remdesivir in dialysis population as below. Also add other papers on covid in special populations

https://pubmed.ncbi.nlm.nih.gov/35203759/

Materials and Methods

Materials and Methods are also very short.

Write about the name of hospital where study was performed.

Results and Discussion

Authors has documented different comorbidities in the study. Do you find any relationship between comorbidities and response to drugs?

Along with MOV and NMV/r, patients were administered other drugs like steroids, anticoagulants?

Write details about the ethical approval, committee name who approved with study, date of approval and letter number.

Author Response

Point 1: Results are the key portion of abstract. Authors describe the vaccination status, adverse effects and age groups in the results. But the true results are the response of MOV and NMV/r. Write in numbers, what is the response of MOV and NMV/r in covid patients with older age.

Response 1: Thank you for your careful review and for pointing this out. We have now added the results on the effectiveness of MOV and NMV/r as primary outcomes in the Abstract.

Point 2: Write a paragraph about the Molnupiravir and nirmatrelvir/ritonavir. What kind of drugs these are? how much effectiveness they showed in clinical trials? what is the response of these drugs in normal adult population?

Response 2: Thank you for these suggestions. Accordingly, we have added a paragraph on the pharmacological mechanism of the drugs and the results of clinical trials among healthy volunteers and real-world settings.

Point 3: Write a paragraph about different drugs, which were used at different stages during the covid pandemic for the treatment of covid. You may include Anticoagulants, Steroids etc.

https://pubmed.ncbi.nlm.nih.gov/34827332/

https://pubmed.ncbi.nlm.nih.gov/34943722/

Response 3: Thank you for this suggestion. We have added a paragraph on the modifications and types of treatment, including corticosteroid and anticoagulant treatment, during the COVID-19 pandemic.

Point 4: Write a paragraph about use of different drugs in covid positive special populations like the older population, population with impaired kidney. You may cite the papers like Remdesivir in dialysis population as below. Also add other papers on covid in special populations

https://pubmed.ncbi.nlm.nih.gov/35203759/

Response 4: Thank you for this suggestion. In the Discussion section, we have described the efficacy of both drugs and the relevant precautions in special populations of COVID-19, such as those with kidney injury, pregnancy, and concomitant drug use.

Point 5: Materials and Methods are also very short.

Write about the name of hospital where study was performed.

Response 5: Thank you for this suggestion. We have now mentioned the hospital name in the Materials and Methods section.

Point 6: Authors has documented different comorbidities in the study. Do you find any relationship between comorbidities and response to drugs?

Response 6: Thank you for this comment. We have added a sentence explaining the relationship between the presence of comorbidities and the patient response to these drugs. However, we could not identify a relationship based on comorbidities.

Point 7: Along with MOV and NMV/r, patients were administered other drugs like steroids, anticoagulants?

Response 7: Thank you for your comment. Only one patient used corticosteroid and anticoagulant medication after admission. None of the other patients used corticosteroid, anticoagulant, and immune modulating drugs during follow-up. We have mentioned this in the Results section in the revised manuscript.

Point 8: Write details about the ethical approval, committee name who approved with study, date of approval and letter number.

Response 8: Thank you for this suggestion. Accordingly, we have added the approval date and number to the Institutional Review Board Statement declaration after the main text.

Reviewer 2 Report

The paper entitled " Real-world experience of the comparative effectiveness and 2 safety of molnupiravir and nirmatrelvir/ritonavir in high-risk 3 patients with COVID-19 in a community setting" is a good and well-designed study. 

It's better if the author could add a diagram/flow chart of the mechanism of COVID-19 and drug action. 

There are a few spelling and English grammar mistakes please correct them. 

Author Response

Point 1: It's better if the author could add a diagram/flow chart of the mechanism of COVID-19 and drug action. 

Response 1: Thank you for this suggestion. Instead of a diagram/flow chart figure, we have added a paragraph on the mechanisms of these drugs for COVID-19 in the Introduction section.

Point 2: There are a few spelling and English grammar mistakes please correct them. 

Response 2: Thank you for this suggestion. We have already used the English proofreading service by Editage (https://www.editage.com/). They have rechecked our manuscript carefully check for grammatical correctness, spelling, and a native tone.

Round 2

Reviewer 1 Report

Manuscript is much improved. Reviewer comments are addressed.

Accept in present form.